# A Novel Continuous-Discontinuous Multi-Field Numerical Model for Rock Blasting

**Yunpeng Li [1], Chun Feng [2,\*], Chenxi Ding [3] and Yiming Zhang [1]**

1. School of Civil and Transportation Engineering, Hebei University of Technology, Tianjin 300401, China
2. Institute of Mechanics, Chinese Academy of Sciences, Beijing 100190, China
3. School of Civil and Resource Engineering, University of Science and Technology Beijing, Beijing 100083, China
* Correspondence: fengchun@imech.ac.cn; Tel.: +86-1381-061-4191

**Abstract:** During blasting, rock failure is caused by blasting wave and explosive gas pressure, as a multi-field coupled process. Most numerical models focus on the effect of blasting wave where the gas pressure is commonly accounted for by empirical relations, ignoring the penetration and permeation of gas flow in cracks. This can underestimate the failure region. In this work, a novel multi-field model is developed in the framework of a continuous-discontinuous element method (CDEM), which is a coupled finite-discrete method with explicit integration strategy. The deformation and cracking of rock mass and the distribution of gas pressure are captured. The proposed method is verified by comparing the results to other results provided in published literature. Especially, by simulating the cases with blocked and unblocked blasting hole, we found that: (i) The fracture degree of the case with blocked blasting hole was 30% higher than that of the unblocked blasting hole. (ii) The radial main cracks in the fracture area are mainly caused by the explosive gas, and the tiny and dense cracks near the hole are induced by the explosion stress wave. (iii) The explosion crushing zone is mainly formed by the action of explosion stress wave, while the crack zone is formed by the combined action of the explosion stress wave and explosive gas. The proposed method provides a useful tool to properly simulate a rock blasting process.

**Keywords:** rock blasting; continuous-discontinuous element method (CDEM); multi-field coupled processes; dynamic loading; gas permeation; numerical simulations

---

## 1. Introduction

Considering rock blasting, the failure of rock mass is induced by blasting wave and successive gas pressure. When the former effect lasts for several microseconds, the latter effect can last for up to 1 millisecond. This whole process is short and accompanied by rapid energy transformation. The effect of blasting wave is widely studied, providing practical empirical formulas for applying equivalent time dependent dynamic loads. However, some empirical equations partly take into account the effect of explosive gas pressure. The gas flow can penetrate and permeate into cracks which promote the formation of main cracks. This process is similar to hydraulic fractures. In other words, ignoring this coupled hydro-mechanical process commonly leads to underestimation of damage regions.

Numerical simulations can assist the researchers to study the rock blasting efficiently and quantitatively. It is difficult for conventional finite element method (FEM) to capture initiations and propagations of cracks of strain-softening materials such as rocks. However, some novel numerical methods are proposed in the framework of continuum mechanics such as numerical manifold methods and extended finite element methods [1–5], cracking element methods (CEMs) [6–10], phase field methods [11–13], cracking particle methods [14–16], and peridynamics [17–21]. Blasting lasts for a very short period, which



commonly results in local fragmentation and large displacements. The hybrid continuous-discontinuous method with explicit integration can be advantageous considering the computing time and numerical stability. For example, the authors of Fakhimi and Lanari [22] simulated a rock blasting process based on a DEM-SPH coupling method, for which the SPH model is used to capture the gas flow and DEM is adopted to simulate rock fractures. The authors of Jayasinghe et al. [23] built an FEM-SPH coupling method to simulate rock blasting and found that the discontinuity and in situ stress of rock have significant influence on the blasting results. The authors of Wang et al. [24] used a hybrid continuous-discontinuous method to simulate the rock blasting induced by high-pressure gas and explosives, indicating that the former case provided fewer radial fractures.

The above research shows that the hybrid continuous-discontinuous method has unique advantages in the field of explosion simulation. In this work, in the framework continuous-discontinuous element method (CDEM) [25–30], we build a coupled hydro-mechanical multi-field model for simulating the rock blasting considering the coupled effects of blasting wave and gas pressure. The main features of our model include:

- The domain is discretized into deformable blocks and interfaces which can capture the continuous-discontinuous process of rock blasting.
- The explosive gas flow is simulated with Darcy's law and plate flow, where the opening of cracks directly increase the gas permeation coefficient.
- A multi-field coupled iterative procedure is designed based on explicit integration framework for assuring efficiency and reliability.

By comparing our numerically-obtained results and experimental results provided by other literatures, our model is validated, which properly predicts the damage region of rock blasting considering blocked and unblocked blasting holes.

The paper is organized as follows: In Section 2, the framework of the continuous-discontinuous element method and multi-field coupling model are introduced, including control equations and numerical procedures. Subsequently, the equivalent loadings are described, including the dynamic loads induced by blasting wave and the quasi-static loads caused by the explosive gas pressure. In Section 3, a classical blasting test is used to verify the model. In addition, the single hole blasting of PMMA(Polymethyl Methacrylate) plate is simulated as the application, comparing to the experimental results considering blocked and unblocked blasting holes. Concluding remarks are given in Section 4.

## 2. Numerical Method

### 2.1. Continuous-Discontinuous Element Method

The continuous-discontinuous element method (CDEM) is a numerical analysis framework with explicit integration built based on the Lagrange equation [31–33]. The elements adopted in this work include deformable block and interface elements, see Figure 1. In Figure 1, the solid black lines represent the real interface such as joints and other initial cracks. The dashed black lines represent the virtual interfaces, which will become activated when damage happens. Normal and tangential springs are arranged in the virtual interface. When these springs reach their strengths, they will break and the interfaces will open. The governing equation of CDEM method is as follows:

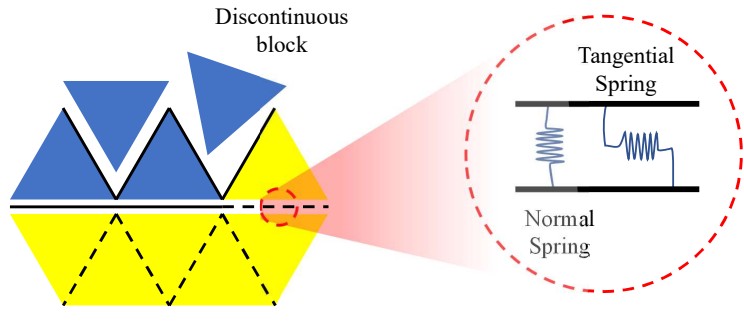

**Figure 1.** Constitutive model in the CDEM.

- Block element
  The governing equation is established based on the Lagrangian energy system.

$$\frac{d}{dt}\left(\frac{\partial L}{\partial \dot{u}_i}\right) - \frac{\partial L}{\partial u_i} = Q_i \tag{1}$$

$$L = \Pi_m + \Pi_e + \Pi_f \tag{2}$$

The energy function of element is

$$L = \frac{1}{2}\int_V \rho \dot{u}_i^2 dV + \int_V \frac{1}{4}\sigma_{ij}(u_{i,j} + u_{j,i})dV - \int_V f_i u_i dV \tag{3}$$

The system external force includes damping and boundary external forces:

$$\begin{cases} Q_\mu = \int_V \mu \dot{u}_i dV \\ Q_{\bar{T}} = -\int_S \bar{T}_i dS \end{cases} \tag{4}$$

Hence, Equation (1) can be rewritten as:

$$-\left(\int_v \rho \ddot{u}_i dV + \int_V \partial_{ij}\frac{\partial_{u_{i,j}}}{\partial_{u_i}}dV - \int_V f_i dV\right) = \int_V \mu \dot{u}_i dV - \int_S \bar{T}_i dS \tag{5}$$

Finally, the block element dynamic function is

$$\mathbf{M}\ddot{u}(t) + \mathbf{C}\dot{u}(t) + \mathbf{K}u(t) = \mathbf{F}(t) \tag{6}$$

- Interface element
  An interface locates between two block elements. The normal and tangential contact forces of the two elements are the spring forces as

$$\begin{cases} F_n = -K_n * \Delta u_n \\ F_\tau = -K_\tau * \Delta u_\tau \end{cases} \tag{7}$$

The relative displacements between adjacent elements are

$$\begin{cases} \Delta u_n = \dfrac{(\sigma_{n1} + \sigma_{n2})A}{2K_n} \\ \Delta u_\tau = \dfrac{(\sigma_{\tau 1} + \sigma_{\tau 2})A}{2K_\tau} \end{cases} \tag{8}$$

The Mohr–Coulomb criterion is adopted; see Equation (9). When the normal stress of the spring is greater than the tensile strength, the normal spring will break and

tensile failure will occur. Similarly, when the tangential spring reaches its strength, the tangential spring will break and shear failure will occur:

$$\begin{cases} F_n = 0, & \sigma_n \geq \sigma_t \\ F_\tau = F_n \tan\phi, & \sigma_\tau \geq c + \sigma_n \tan\phi \end{cases} \tag{9}$$

### 2.2. Permeation of Gas in Cracks

For obtaining the distribution of gas pressure, the finite volume method is used. When the springs connecting the adjacent elements break, Darcy's law and plate flow are considered to simulate the gas permeation in these cracks. The velocity of gas flow $V$ is obtained with

$$V = -k_s \frac{\partial P_g}{\partial x} \tag{10}$$

where $k_s$ is the gas permeation coefficient in crack. Considering the plate flow and cubic low, we have

$$k_s = \frac{\omega^2}{12\mu_g} \tag{11}$$

where $\omega$ is the normal crack opening, and $\mu_g$ is the viscosity of gas.

An iterative procedure is designed to trace the gas flow in the cracks. When the crack opens, the gas pressure of the nodes on the two crack surfaces is updated. The gas flow permeates further and results in increasing gas pressure in the deeper region. This iterative algorithm is shown in Figure 2.

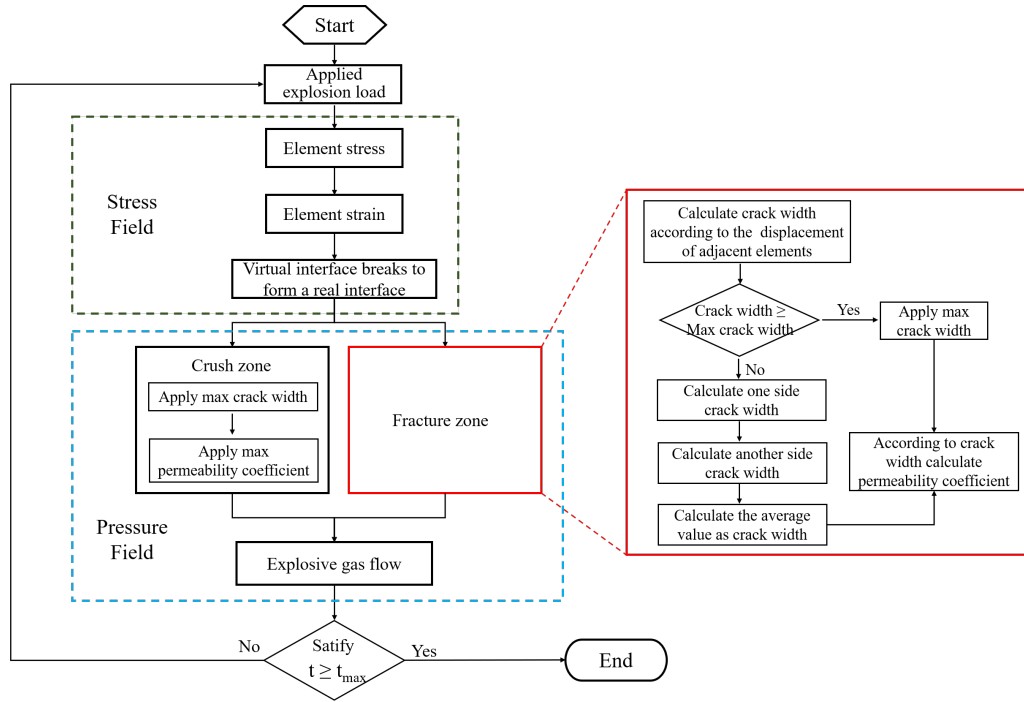

**Figure 2.** The iterative procedure for calculating the coupled deformation/pressure fields of cracked rock mass.

### 2.3. Explosion Load

Yuan et al. [34] suggests a time-dependent explosion pressure curve with two peaks, see Figure 3a. The first peak corresponds to the blasting wave, and the second peak corresponds to the gas pressure. As mentioned before, comparing to the explosion stress wave which acts as dynamic loading, the explosive gas permeation process lasts longer, which is considered as a quasi-static loading.

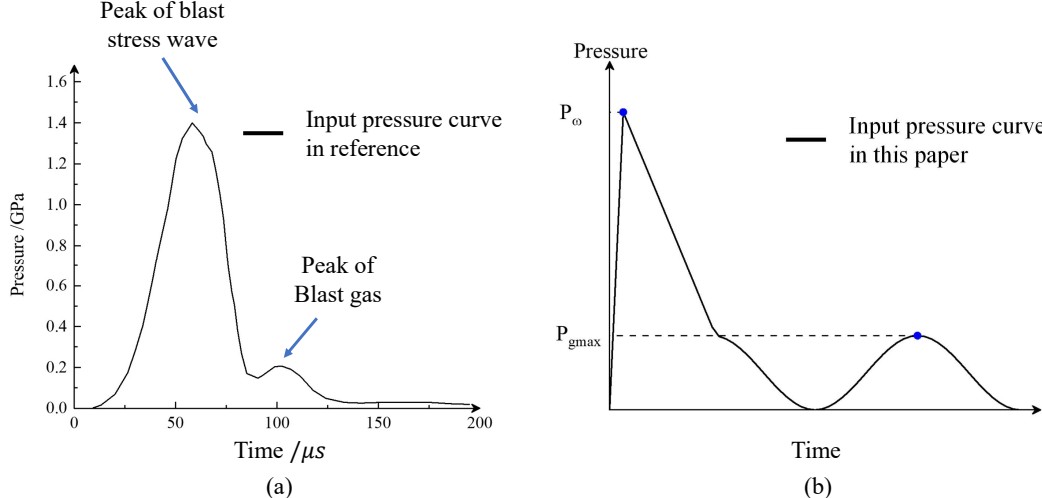

**Figure 3.** Explosion load with the form of dynamic and quasi-static: (**a**) pressure curve suggested in [34]; (**b**) pressure curve used in this paper.

The first peak pressure $P_w$ can be determined base on C–J detonation theory, which is

$$P_w = \frac{\rho_0 \, D^2}{2(\gamma + 1)}. \tag{12}$$

Furthermore, the action time $t_w$

$$t_w = \frac{r_s}{V_{cp}}. \tag{13}$$

According to [35], time of blast pressure ascent stage $t_{wr} = 0.1 \, t_w$ and descent stage $t_{wf} = t_w - t_{wr}$.

The second peak pressure of explosive gas $P_{gmax}$ was calculated based on Park et al. [36]:

$$P_{gmax} = \begin{cases} \dfrac{\rho_0 D^2}{8}(\zeta)^{-6}, & P_d \le P_k \\[2ex] P_k \left( \dfrac{\rho_0 D^2}{1.6 \times 10^9} \right)^{0.467} (\zeta)^{-2.8}, & P_d > P_k \end{cases} \tag{14}$$

where $\zeta$ is the uncoupling coefficient of blasting hole, $\zeta = r_b \, / \, r_e$ where $r_b$ is the radius of hole, and $r_e$ is the radius of charge. Based on [37], we assume a sine curve for the second peak. The complete time dependent explosion pressure curve used in this work is shown in Figure 3b.

A personal computer with an AMD Ryzen Threadripper 3960X 3.90 GHz processor and 32 GB memory was used to obtain the numerical results. A benchmark model with 10,000 triangular elements will take 2 h, and a complex case such as Section 3.2 of this paper (with 21,896 triangular elements and 32,972 interface elements) will take about 6 h.

## 3. Numerical Example

### 3.1. Single Hole Blasting Test

To verify the proposed model, the single hole blasting test is considered. The model is a granite disk with radius 72 mm and a hole with a radius 3.225 mm, which is filled with PETN (Pentaerythritol Tetranitrate) [38], as a plane stress condition. The material parameters are taken based on [39]; see Table 1 and the parameter for explosive is listed in Table 2. In total, 39,480 triangular elements and 59,672 interface elements are created in the numerical model.

**Table 1.** Granite model main parameters [39].

| Material Properties | Units | Value |
|---|---|---|
| Density $\rho$ | (kg/m$^3$) | 2660 |
| Modulus of elasticity $E$ | (GPa) | 51 |
| Poisson's ratio $\mu$ | (-) | 0.16 |
| Cohesive force $C$ | (MPa) | 25 |
| Tensile strength $f_t$ | (MPa) | 7.3 |
| Angle of internal friction | ° | 56.4 |
| The dilatancy angle | ° | 28.2 |

**Table 2.** PETN main paremeters [38].

| Material Properties | Value |
|---|---|
| Charge density $\rho_w$ (kg/m$^3$) | 1770 |
| Specific internal energy of explosive U (J/m$^3$) | $1.010 \times 10^{10}$ |
| C–J pressure $P_{CJ}$ (GPa) | 3.2 |
| Detonation velocity V (m/s) | 8300 |

The obtained cracking pattern is shown in Figure 4, compared to the MPM-CDEM results provided by [39] and experimental results provided in [38]. Figure 4 indicates that the explosion pulverized formed a pulverized zone around the blasting hole, and some radial cracks propagate to the edge of the model. Close to the edge of the model, some tangential cracks appear due to stress wave reflection, resulting in a nearly circular fractured region. Assuming the pulverized zone as the blasting funnel, the radius of the blasting funnel and number of main cracks are listed in Table 3, indicating the reliability of our model.

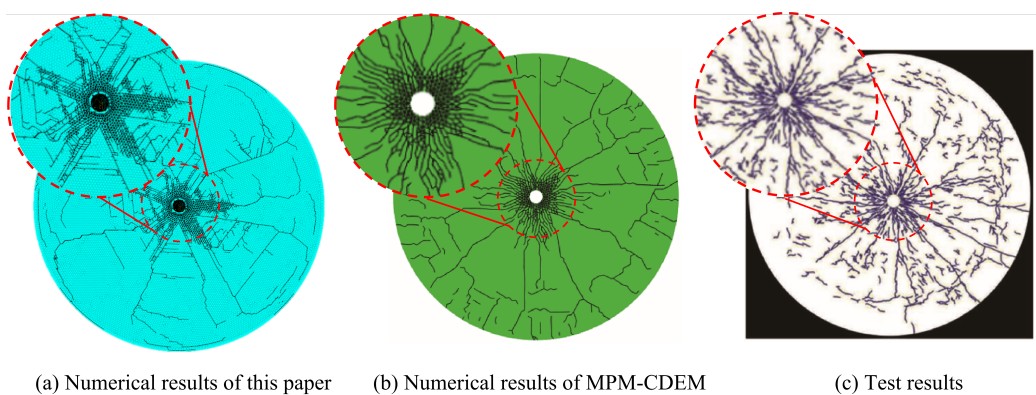

(a) Numerical results of this paper     (b) Numerical results of MPM-CDEM     (c) Test results

**Figure 4.** Cracking pattern of the single-hole blasting test.

**Table 3.** The radius of the blasting funnel and number of main cracks.

| | This Paper | MPM-CDEM [39] | Test [38] |
|---|---|---|---|
| Radius of blasting funnel (mm) | 8.51 | 7.85 | 9.81 |
| Number of main cracks (-) | 11 | 14 | 12 |

### 3.2. Blasting of the PMMA Plate

We used the proposed method to simulate the blasting test of the PMMA plate [40], see Figure 5. This case is also plane stress. We used Gmesh software to build a two-dimensional square model with a side length of 300 mm and set a blasting hole with a radius of 2 mm in the center of the model. We divided the model into 21,896 triangular elements and 32,972 interface elements, and the size of elements is 1–5 mm. To prevent the explosive

stress wave reflection, we set a non-reflective boundary condition around the model. The material properties are listed in Table 4, which is determined based on [41]. The explosive is $PbN_3$, the properties of which are listed in Table 5.

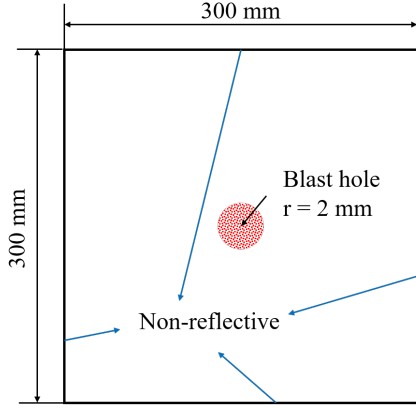
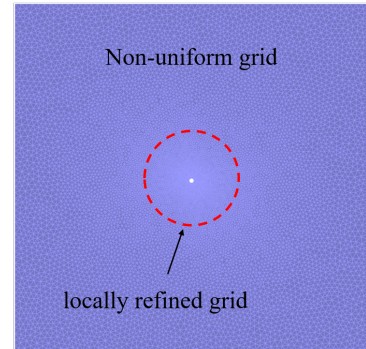

**Figure 5.** Numerical model.

**Table 4.** PMMA model main parameters.

| Material Properties | Value |
| --- | --- |
| Density $\rho$ (kg/m$^3$) | 1180 |
| Modulus of elasticity E (GPa) | 6.1 |
| Poisson's ratio $\mu$ (-) | 0.31 |
| Cohesive force C (MPa) | 66 |
| Tensile strength T (MPa) | 66 |
| Angle of internal friction $^\circ$ | 35 |
| The dilatancy Angle $^\circ$ | 5 |

**Table 5.** $PbN_3$ main parameters.

| Material Properties | Value |
| --- | --- |
| Charge density $\rho_w$ (kg/m$^3$) | 2560 |
| Explosion heat Q (kJ/kg) | 1524 |
| C–J pressure $P_{CJ}$ (GPa) | 3.59 |
| Detonation velocity V (m/s) | 2250 |

According to [40], two conditions are considered as (i) blocked blasting hole and the (ii) unblocked blasting hole, where the gas permeation in the second condition can be ignored in the experiment. The obtained results are shown in Figure 6.

Figure 6a,b show the numerically-obtained results, and Figure 6c,d show the experimental results. It can be seen that, when the hole is not blocked, there is almost no radial main crack. Only dense cracks are generated near the hole, resulting in a crush zone caused by the blasting wave. When the hole is blocked, several radial main cracks are generated, which are caused by the gas permeation. The numerical results agree well with the experimental results, indicating the reliability of the proposed model. We would like to emphasize that we did not use special techniques to simulate explosion crushing area such as element dissolution/deletion because such techniques do not follow mass and energy conservation, which may bring other problems. Hence, in our results, a large number of micro-cracks appeared on the edge of the explosion crushing area in Figure 6.

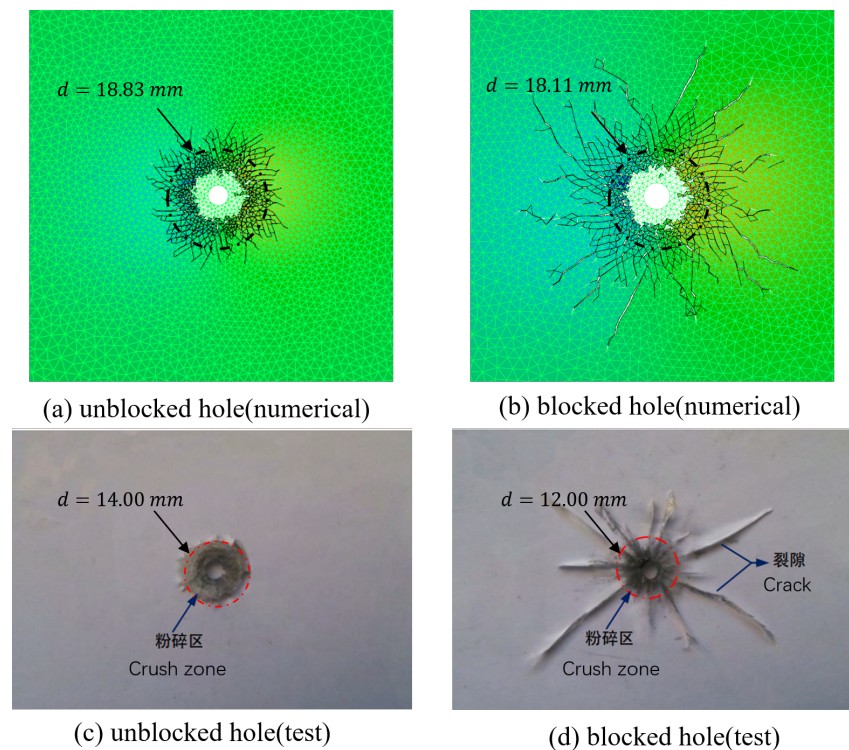

(a) unblocked hole(numerical)

(b) blocked hole(numerical)

(c) unblocked hole(test)

(d) blocked hole(test)

**Figure 6.** Cracking pattern.

Considering the blocked condition, the evolution of gas pressure is shown in Figure 7.

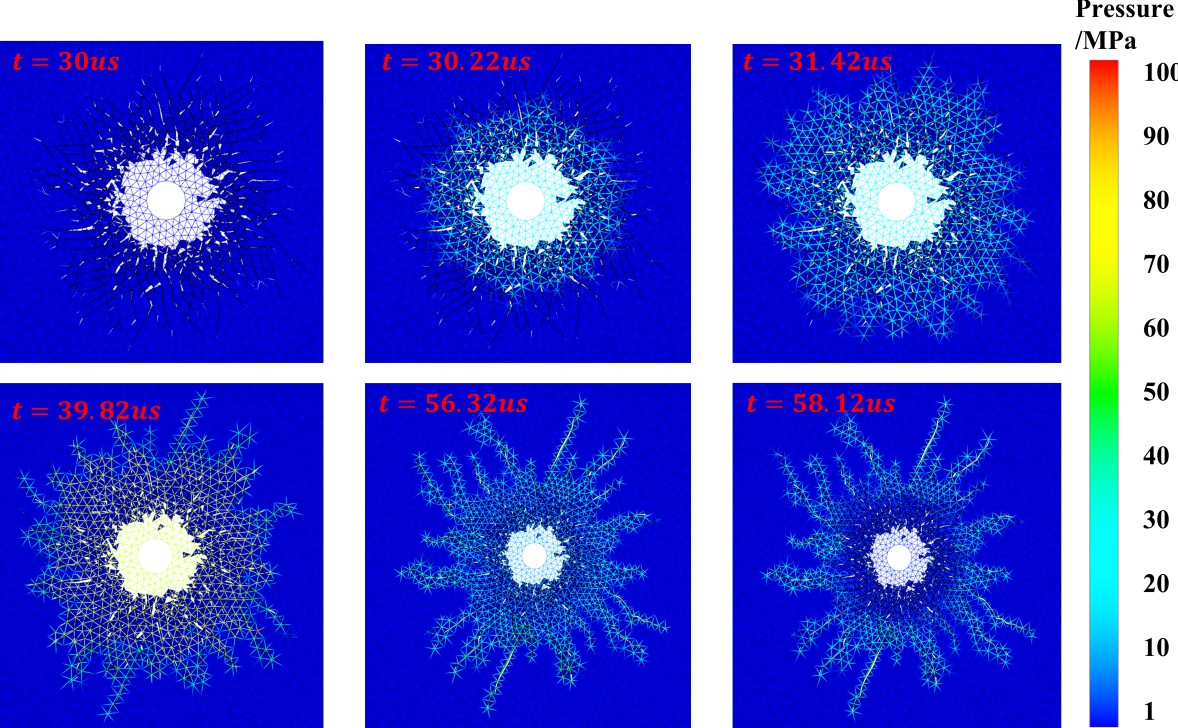

**Figure 7.** Pressure of explosive gas.

The damage region at $t = 30$ μs in Figure 7 is similar to the final damage region of the case with an unblocked blasting hole, indicating that the effect of blasting wave has finished. At $t = 30.22$ μs, the explosive gas begins to permeate into the crush zone with

maximum gas pressure around 20 MPa. When $t = 39.82$ µs, the maximum gas pressure exceeds the strength of the PMMA plate. At $t = 56.32$ µs, the main cracks appear, and the gas begins to escape. The maximum gas pressure also drops. Finally, at $t = 58.12$ µs, the cracks become smooth, and the whole process ends.

The results of Mises stress are shown in Figure 8. In the beginning of the explosion, the blasting hole area was subjected to strong impacts where the stress peak reached 100 MPa. Before 30 µs, the explosive stress wave spread almost symmetrically. It can be found that the blasting wave almost reached the boundary of this region in 30 µs. With the propagating of the explosion wave, crushing and cracking zones were formed. When $t > 30$ µs, the explosive gas permeated into cracks, promoting the cracking processes. At $t = 36.82$ µs, the cracked zone was almost formed where the stress concentrations could be found at the tips of cracks. At $t = 60$ µs, the explosive gas began to dissipate. (The blue grid is the fracture grid which supports the explosive gas flow in the crack).

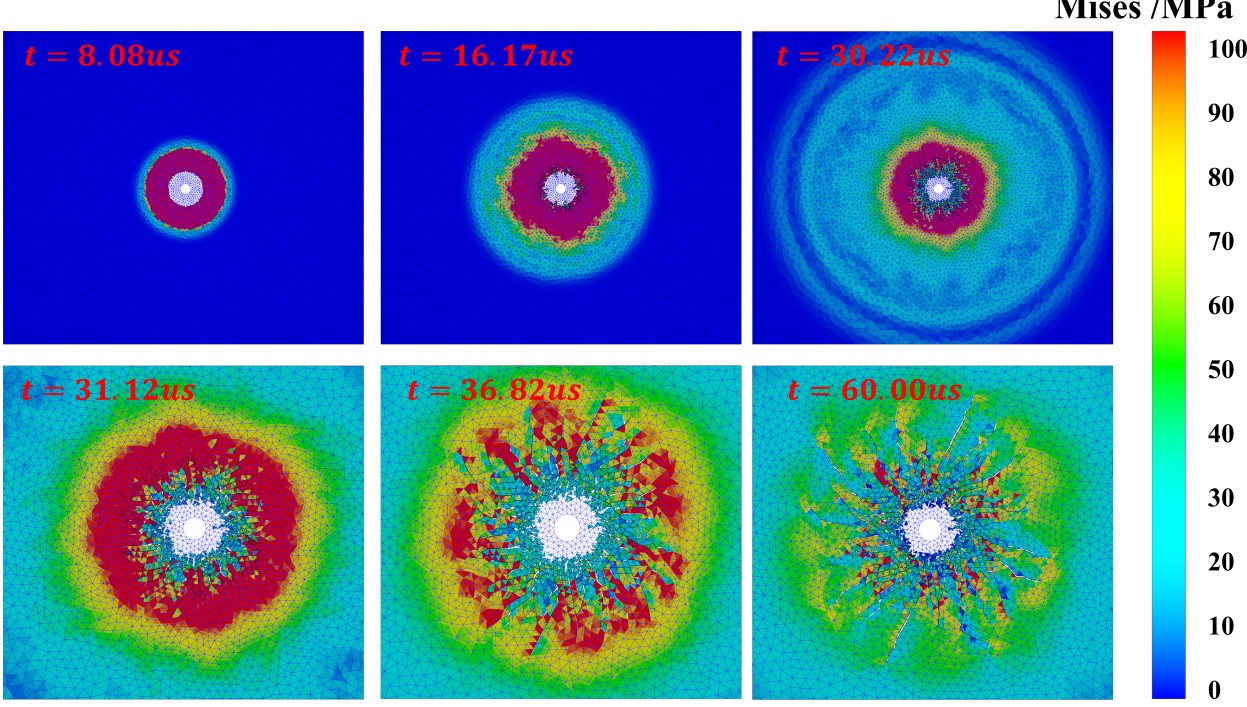

**Figure 8.** Mises stress of the PMMA plate.

When defining the ratio of the number of broken springs to the total number of springs as fracture degree, the history of fracture degree is shown in Figure 9. It can be seen that, in the first 30 µs, the fracture degree of the cases with blocked and unblocked blasting holes are almost the same. After this, the fracture degree of the case with a blocked hole continuously increases, which is 30% higher than the case with the unblocked blasting hole in the final stage. This conclusion is consistent with Ref. [40], and it also verifies the proposed model's rationality. However, due to the calculation efficiency, the proposed model is only suitable for the two-dimensional case, and the three-dimensional case will take too much time in calculating the explosive gas flow in the crack. This is also our future work.

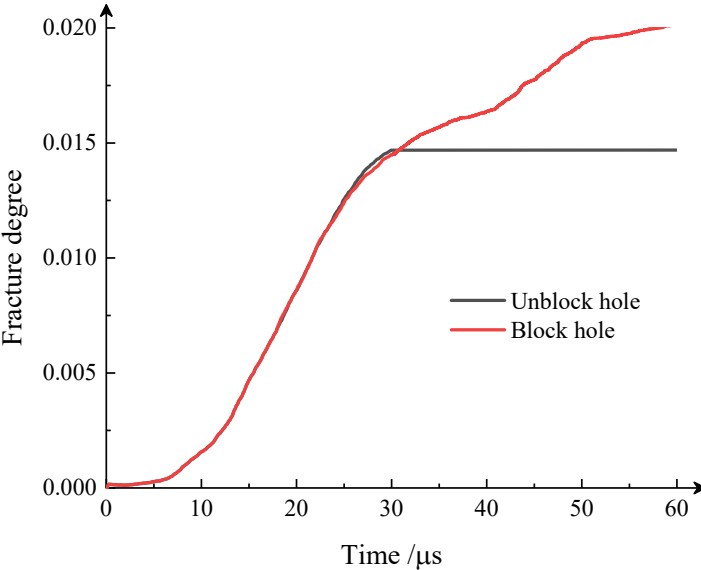

**Figure 9.** Fracture degree.

## 4. Conclusions

A coupling method is proposed to effectively simulate explosive cracks in rock. The plate flow equation is used to describe the process of gas expansion and pressure propagation in explosion; CDEM is used to describe the process of blast stress wave shocking and crushing rock. By combining the dynamic and quasi-static loads, the progressive failure process of rock from continuous to discontinuous is reproduced. Compared to previous studies, the role of explosive gas in the explosion is considered, providing more reliable results of damage regions. The effectiveness and accuracy of the method in blasting are proven by numerical examples. The main conclusions of this paper are as follows:

1. The whole process of rock blasting including the blasting stress wave, explosive gas permeation, and fracking processes are reproduced by the proposed model.
2. For single-hole PMMA plate blasting, the established numerical model captures the crush and fracture zones. By comparing the two cases with blocked and unblocked blasting holes, it is found that the crushing zone in the explosion is mainly caused by the explosion stress wave. The tiny and dense cracks in edge of the fracture zone are caused by the reflection of the stress wave at the boundary. Meanwhile, the radial main cracks developed in the specimens are caused by the explosive gas. By comparing the fracture degrees, it can be found that the explosive gas can account for around 30% of the total damage.
3. The explosion crushing zone is mainly formed by the action of the explosion stress wave, while the crack zone is formed by the combined action of the explosion stress wave and explosive gas.

**Author Contributions:** Conceptualization, Y.L.; literature search, Y.L.; Funding acquisition, C.F. and Y.Z.; method feasibility analysis, Y.L. and C.F.; algorithm development, Y.L.; validation, Y.L. and C.D.; Writing—original draft, Y.L.; Writing—review and editing, Y.L. and Y.Z. All authors have read and agreed to the published version of the manuscript.

**Funding:** This research was funded by the National Natural Science Foundation of China (Grant No. 52178324), the National Key Research and Development Project of China, and the Ministry of Science and Technology of China (Project No. 2018YFC1505504). The APC was funded by Y.M. Zhang.

**Institutional Review Board Statement:** Not applicable.

**Informed Consent Statement:** Not applicable.

**Data Availability Statement:** The data used to support the findings of this study are included within the article.

**Acknowledgments:** The authors would like to acknowledge the financial support of the National Natural Science Foundation of China (Grant No. 52178324), the National Key Research and Development Project of China, and the Ministry of Science and Technology of China (Project No. 2018YFC1505504).

**Conflicts of Interest:** The authors declare no conflict of interest.

## Abbreviations

List of symbols

| | | |
|---|---|---|
| $Q_i$ | (J) | The work of external force |
| $L$ | (-) | Lagrangian function |
| $\ddot{u}_i$ | (m/s$^2$) | Generalized acceleration |
| $\dot{u}_i$ | (m/s) | Generalized velocity |
| $u_i$ | (m) | Generalized displacement |
| $t$ | (s) | Time |
| $\Pi_e, \Pi_f, \Pi_m$ | (J) | System kinetic energy, elastic energy, potential energy |
| $\rho, \rho_0$ | (kg/m$^{-3}$) | Density of material, charge |
| $\sigma_{ij}$ | (-) | The mean strain tensor of the element |
| $V$ | (m$^3$) | Volume of element |
| $\mu$ | (N/(m/s)) | Damping |
| $\bar{T}_i$ | (N/m$^2$) | The surface force on the element |
| **M** | (-) | Element mass stiffness matrix |
| **C** | (-) | Damping matrix |
| **K** | (-) | Stiffness matrix |
| **F**$(t)$ | (-) | External load array of nodes |
| $F_n, F_\tau$ | (N) | Normal,Tangential contact force of adjacent elements |
| $K_n, K_\tau$ | (-) | Normal,Tangential stiffness of the spring |
| $\Delta u_n, \Delta u_\tau$ | (m) | Normal,Tangential displacement of adjacent elements |
| $\sigma_{n1}, \sigma_{n2}$ | (MPa) | Normal stress of contact point pair |
| $\sigma_{\tau 1}, \sigma_{\tau 2}$ | (MPa) | Tangential stress of contact point pair |
| $A$ | (m$^2$) | The area of the contact point pair |
| $c$ | (N) | Cohesive force |
| $\phi$ | (°) | Internal friction angle |
| $V_i$ | (m/s) | Velocity of node explosive gas flow |
| $P_g$ | (MPa/m) | Pressure gradient of explosive gas |
| $x_i$ | (-) | The coordinate component of the direction i of a node |
| $k_s$ | (m$^2$/Pa·s) | Gas permeability coefficient |
| $\omega$ | (m) | The width of crack |
| $\mu_g$ | (Pa·s) | The dynamic viscosity of fluid |
| $P_w$ | (MPa) | The peak pressure of explosive |
| $D$ | (m/s) | Detonation velocity |
| $\gamma$ | (-) | Gas adiabatic index |
| $t_w$ | (s) | Blast stress wave action time |
| $r_s$ | (m) | The radius of pulverized zone |
| $V_{cp}$ | (m/s) | Crack propagation velocity under shock wave |
| $P_k$ | (MPa) | Critical pressure, $P_k$ = 200 MPa is considered in this work |

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
