# Peer review of "A Novel Continuous-Discontinuous Multi-Field Numerical Model for Rock Blasting"

_applsci, doi:10.3390/app122111123_

Round 1
Reviewer 1 Report
This manuscript presents a multifield model constructed with the continuous-discontinuous element method (coupled model) with an integration strategy. The manuscript needs some improvement before it can be processed further. See my comments attached.

Author Response
Dear Reviewer:
According to the comments,we have responded point-by-point and revised our manuscript. Please see the attachment.

Reviewer 2 Report
Although the scientific topic of the paper is interesting, the content of the paper, research design, the results and the conclusion section show that this manuscript is suitable to be considered as a report or a conference paper. Besides, the paper is written in poor English so it must be considerably improved. Some of the important papers referenced are not easily accessible to researchers because they are not in English. Some comments and suggestions for authors are reported as follows:
1. In line 25, the sentence “In experimental investigations researchers have found ...” is from which references?
2. In the last paragraph of the introduction, there is no need to refer to the section numbers.
3. In line 76, what do the references refer to? It does not seem necessary.
4. On page 5, Equation 9 shows the Mohr-Coulomb criterion, not Equation 14. “The Mohr-Coulomb criterion is adopted, see Equation 14.”
5. Parameters A, B, R1, R2 and ω in Table 2 are not defined in the text. What are these parameters?
6. What words are “PMMA” and “PETN” in lines 71 and 109 respectively abbreviated?
7. Comparing the numerical and experimental results in Figure 6 shows a 35 to 50% difference between the results and no good agreement is observed. What is the reason for this unacceptable difference? The authors should mention to this difference in the paper and correct the numerical model.
8. The tests used for validation were not described in the paper.
9. Presentation of results, analysis of results and conclusion section are not appropriate for a research paper and should be revised.
10. The English language of the manuscript needs to be improved. Some of the grammatical errors and typos are mentioned below:
Line 2: “resulted by blasting”: resulted from blasting
Line 4: “underestimates”: underestimate
Line 26: “bigger of the cases”: bigger in the cases
Line 28, 71, 112 and 154: “Comparing to”: Compared to
Lines 29: “It is difficulty”: It is difficult
Line 35: “results into”: results in
Line 41: “The thought that hybrid”: The thought was that the hybrid
Line 67: “the framework … are introduced”: the framework … is introduced
Line 156: “results of damage region”: results for the damaged regions
Author Response

(The authors gave the same response as above.)

Round 2
Reviewer 2 Report
The authors responded to all the comments from my previous review.